# *Lonicera caerulea* Pomace Alleviates DSS-Induced Colitis via Intestinal Barrier Improvement and Gut Microbiota Modulation

**DOI:** 10.3390/foods12183329

**Published:** 2023-09-05

**Authors:** Baixi Zhang, Xinwen Huang, Lijuan Niu, Xuemei Chen, Bo Hu, Xiaoshu Tang

**Affiliations:** 1School of Food Science and Technology, Jiangnan University, Wuxi 214122, China; 6200112030@stu.jiangnan.edu.cn (X.H.); 6200113070@stu.jiangnan.edu.cn (L.N.); chenxm@jiangnan.edu.cn (X.C.); hubo@jiangnan.edu.cn (B.H.); txs@jiangnan.edu.cn (X.T.); 2National Engineering Research Center for Functional Food, Jiangnan University, Wuxi 214122, China

**Keywords:** *Lonicera caerulea*, ulcerative colitis, anti-inflammatory, intestinal barrier, gut microbiota

## Abstract

The objective of this investigation was to appraise the mitigative effects of *Lonicera caerulea* pomace (LCP)-supplemented diets on Dextran Sulfate Sodium (DSS)-induced colitis, and to discuss the potential mechanisms. LCP, a by-product of fruit juice processing, harbors a higher content of polyphenols and dietary fiber compared to the *L. caerulea* pulp. In a murine model of colitis, the LCP diet attenuated the symptoms of colitis, as evidenced by the reduction in the disease activity index (DAI), extension of colon length, and amelioration of histopathological damage. The anti-inflammatory attributes of LCP were substantiated by a decrease in myeloperoxidase (MPO) activity and suppression of inflammatory cytokine expressions within the colon. Meanwhile, LCP mediated the repair of the intestinal barrier, characterized by the upregulation of gene expressions of tight junction (TJ) proteins and *Muc2*. Furthermore, LCP altered the composition of the gut microbiota, manifested in increased alpha diversity, enhancement of the abundance of beneficial bacteria (*Akkermansia*, *Coprococcus* and *Bifidobacterium*), and diminishment in the abundance of pathogenic bacteria (*Escherichia*, *Enterococcus*, *Mucispirillum* and *Clostridium*). Dietary LCP also increased the concentrations of SCFAs within the intestinal luminal contents of colitis mice. Given the affirmative impact of LCP on colitis, LCP may possess great potential in promoting intestinal health.

## 1. Introduction

Crohn’s disease and ulcerative colitis (UC) are two types of inflammatory bowel disease (IBD), both of which are rising in prevalence and incidence globally. The highest prevalence was recorded in North America and Europe, with newly industrialized nations confronting a surge in incidence [1]. Although the exact mechanism of IBD remains unknown, it is commonly considered that the pathophysiology involves gene inheritance, immune imbalance, and host responses to exogenous agents [2]. Typical clinical manifestations of IBD include loss of appetite, abdominal pain, bloody diarrhea, and mucus discharge [3]. Current therapeutic modalities such as Mesalazine, glucocorticoids, and immunomodulators, though effective, entail considerable financial burden and are associated with serious adverse events including pancreatitis and renal toxicity, necessitating vigilant patient monitoring [4,5]. Consequently, there emerges a need to explore innovative preventative and therapeutic approaches leveraging natural substances and dietary supplements. In the DSS model, prior investigations have revealed a marked variance in survival rates, reflecting a 6.3-fold differential between the most detrimental (4.4 days) and beneficial (27.6 days) dietary regimens, underscoring the influential role of diet in colitis management [6]. Accumulating studies indicates the potential of specific fruits and vegetables, such as berries, to attenuate colonic inflammation [7,8], with polyphenols identified as crucial contributors [9,10]. These bioactive compounds are generally characterized by anti-inflammatory, antioxidant, immune-regulatory, and apoptotic modulatory functions [11], with their colonic metabolites exhibiting effective absorption properties [12]. These metabolites have the potential to mitigate the presence of free radicals in the colon via scavenging mechanisms [13]. The dietary fiber inherent in berries, upon fermentation by gut microbiota, leads to the production of short-chain fatty acids (SCFAs) in the colon [14], contributing to the regulation of intestinal inflammation [15] and reinforcement of the intestinal barrier [16].

*Lonicera caerulea*, also known as honeysuckle berry or haskap, belonging to the Caprifoliaceae family, is well-adapted to cool and humid climes and boasts exceptional cold resistance. Renowned for its rich profile of bioactive metabolites, it is known as a “homology of medicine and food”. In December 2018, according to European Union standards, *L. caerulea* received recognition as a traditional food from a third world country [17]. Predominantly distributed in frigid zones such as Heilongjiang, the Jilin province of China, Japan, and Russia, *L. caerulea* is rich in phenolic compounds, anthocyanins, organic acids, Vitamin C, and minerals [18,19], demonstrating potential in reducing triglyceride absorption, modulating gut microbiota to prevent obesity [20], managing sarcopenic obesity via fat reduction and muscle-related gene regulation [21], mitigating inflammation in non-alcoholic fatty liver disease [22], and positively altering the intestinal environment [23]. *L. caerulea* pomace (LCP), a by-product of juice processing, is notably rich in dietary fiber and polyphenols. Jan Oszmiański et al., reported that peel-based dried pomace of *L. caerulea* harbored a greater quantity of polyphenolic compounds than its pulp [19]. However, most non-extractable polyphenols (NEPs) were conjoined with macromolecular substances such as dietary fiber in fruit pomace, rendering them inaccessible to extraction via organic solvents due to challenges in both extraction and analysis. Although these NEPs and macromolecules are not directly absorbed by the human body, evidence suggests that they would reach the colon and undergo decomposition by gut microbiota, showing antioxidant activity and yielding absorbable metabolites [24]. Such attributes posit LCP as a potential agent in maintaining intestinal homeostasis and inhibiting inflammation. Given the limited existing research on LCP’s salutary impacts on intestinal health, it becomes imperative to undertake an in-depth examination of LCP-supplemented dietary intervention for colitis. In the present study, we have repurposed LCP and assessed the efficacy of LCP supplementation in DSS-induced colitis in mice.

## 2. Materials and Methods

### 2.1. Chemical and Reagents

A Plant Total Phenol (TP) Content Assay Kit was obtained from Sangon Biotech Co., Ltd. (Shanghai, China). DSS (molecular weight: 36,000–50,000 Da) was acquired from MP Biomedicals (Irvine, CA, USA). The commercial kit for myeloperoxidase (MPO) analysis was obtained from Nanjing Jiancheng Bioengineering Institute (Nanjing, China), and the enzyme-linked immunosorbent assay (ELISA) kit for determination of tumor necrosis factor-α (TNF-α) was obtained from Shanghai Enzymelinked Biological Technology Co., Ltd. (Shanghai, China). All other chemicals employed were of commercially accessible analytical grade.

### 2.2. LCP Preparation and Supplemental Dose Evidence

*L. caerulea* berries, of the variety ‘Beilei’, were obtained from FengRan Agricultural Group Company, Heilongjiang, China. The juice extractor (VP21E) from Slovenia was utilized to press the fruit mass without water addition, and the resultant pomace was collected and stored at −16 °C for further processing. AIN-93M diets were supplemented with 5% and 10% LCP by weight and manufactured by Trophic Animal Feed High-Tech Co., Ltd. (Nantong, China). The diet formula was modified to ensure the caloric composition of the LCP diet was congruent with that of the control diet (Table 1). Ash, lipids, protein, sugars (fructose, glucose, sucrose, and maltose), and dietary fiber of LCP were analyzed by dry-ashing, acid-hydrolysis/ether extraction, Kjeldahl, HPLC, and Prosky methods, respectively [25,26,27,28,29]. The supplemental dose of LCP was determined in accordance with the quantity of grape pomace and other pomaces bearing a similar composition to LCP employed in previous colitis mitigation experiments [16,30,31]. The dose of 5% LCP supplement equated to 50 g/kg of the diet. The average dietary intake of a mouse weighing 22 g was 3 g per day, which equals 150 mg LCP per day (6.82 g LCP/day/kg body weight). The figure of 6.82 LCP/day/kg bw for a mouse converts to 33.17 g of LCP dietary intake per day for a 60 kg human, utilizing the body-surface-area normalization methods [32].

### 2.3. Animal Experimental Design

Male C57BL6/J mice (22 ± 1 g, 6 weeks) were purchased from GemPharmatech (Beijing, China). These animals were accommodated in a specific pathogen-free environment at the Laboratory Animal Center of Jiangnan University (license no. SYXK (Su) 2021-0056, Wuxi, China) under controlled humidity (40–70%) and temperature (20–26 °C), with a 12-h light/dark cycle. Jiangnan University Experimental Animal Management and Animal Welfare Ethics Committee reviewed and approved the animal study (JN. No. 20211215c0600128 [532]). The experimental protocols were as follows (Figure 1A): mice were acclimatized for one week with sterile water and standard diets (AIN-93M) prior to being randomly assigned to four groups (eight mice/group): (1) control group fed with AIN-93M diet (C), (2) colitis model group fed with AIN-93M diet (M), (3) AIN-93M diet supplemented with 5% LCP (P5), and (4) AIN-93M diet supplemented with 10% LCP (P10). On days 18–25, the drinking water of all groups, except the control group, was replaced with a 2.5% (*w*/*v*) DSS aqueous solution to induce colitis [33]. Fecal samples were collected on day 25, and the mice were subsequently sacrificed following a 12-h fast. Colon length was measured before collection into sterile tubes. Organ indexes were calculated using the formula: organ index = organ weight (g)/body weight (g) × 100%. Colon tissues were promptly frozen in liquid nitrogen following the retrieval of 1 cm tissues for hematoxylin and eosin (H&E) staining.

### 2.4. Myeloperoxidase (MPO) and Enzyme-Linked Immunosorbent Assay (ELISA)

Colon tissue was precisely weighed, and a 5% tissue homogenate was prepared through the addition of precooled PBS (0.01 M, pH = 7.4) at a ratio of 1:19 (*w*/*v*). The tissue homogenate was subsequently homogenized with distilled water and chromogenic solution, followed by incubation in a water bath at 37 °C for 30 min. The reaction was terminated with a 10-min bath at 60 °C after stop solution was added. Absorbance was measured immediately, using a 460 nm wavelength and a 1 cm optical path. A unit of MPO activity is described as the amount of 1 mol H_2_O_2_ degraded at 37 °C, with the value represented in units per gram (U/g).

After the protocol for the ELISA kit, 50 μL of stop solution was added to the serum solution, and the absorbance at a 450 nm wavelength was measured immediately to determine the concentration of TNF-α in the mouse serum.

### 2.5. Alcian Blue and Periodic Acid–Schiff (AB-PAS) Staining

Alcian blue staining was employed to analyze the distribution of colonic mucin according to the method described by Steedman [34]. Periodic acid–Schiff staining was used to determine goblet cells. The experiment was conducted by Superbiotech Company (Nanjing, China).

### 2.6. Real-Time Quantitative Polymerase Chain Reaction (RT-qPCR)

Total RNA was isolated from colon tissues utilizing the FastPure^®^ Cell/Tissue Total RNA Isolation Kit V2, and the purity and integrity were tested by measuring the optical density at 260/280 nm. The HiScript^®^ III All-in-one RT SuperMix (Vazyme, Nanjing, China) was used to synthesize cDNAs, and the CFX96 Real-Time PCR System (Bio-Rad, Hercules, CA, USA) facilitated the RT-qPCR utilizing SYBR Green qPCR Master Mix (Vazyme, China). The PCR amplification procedure was as follows: predenaturing at 95 °C for 30 s; 95 °C for 10 s; 60 °C for 30 s, and 40 cycles, with the instrument’s default melting curve acquisition program. Primers were synthesized by Sangon Biotech Co., Ltd. (Shanghai, China). Table 2 presents the primer sequences. Gene expression levels were normalized to the *β-actin* gene, employing the 2^−ΔΔCt^ method.

### 2.7. Determination of SCFAs in the Intestinal Contents

Colon contents (0.05 g) were resuspended in 500 μL of precooled saturated sodium chloride solution and subjected to an ice bath for 30 min. A 20 μL aliquot of 10% (*v*/*v*) sulfuric acid solution was used to acidify the colon contents. Following 30 s of vortexing, 1 mL of anhydrous ethyl ether (containing 1 mmol/L of internal standard 2-ethylbutyric acid) was added for fatty acid extraction. After vortexing for 30 s, centrifugation was performed at 4 °C and 18,000× *g* for 15 min, and the supernatant was dehydrated with 0.25 g of anhydrous sodium sulfate, standing in an ice bath for 30 min. Post-centrifugation, the supernatant was filtered through a 0.22 μm membrane filter for detection. Gas chromatography was utilized to assess the levels of acetic, propionic, butyric, isobutyric, valeric and isovaleric acids in the intestinal contents. Instrument parameters: DB-FFAP capillary column (30 m × 0.25 mm × 0.25 μm), initial temperature of 50 °C for 1 min, rising to 120 °C at 15 °C/min, held for 5 min, and then increased to 250 °C for 1 min. The hydrogen flame ionization detector temperature was set at 255 °C, with nitrogen flowing at 1.5 mL/min; the sample dosage was 2 μL, with a split ratio of 10:1. Calibration data utilized 2-ethylbutyric acid as an internal standard. All standards were purchased from Macklin Biochemical Co., Ltd. (Shanghai, China).

### 2.8. Fecal Microbial Gene Sequencing and Analysis

Following 7 days of DSS administration, fecal samples were collected for genomic DNA extraction. The PCR reaction system was configured using 30 ng of qualified genomic DNA and primers specific to the V3-V4 region, with the primers being 341F (5′-ACTCCTACGGGAGGCAGCAG-3′) and 806R (5′-GGACTACHVGGGTWTCTAAT-3′). PCR products were subsequently purified, dissolved by Agencourt AMPure XP (Beckman Coulter, CA, USA), and tagged to enable library construction. Qualified libraries were sequenced on a HiSeq platform from BGI Genomics (Shenzhen, China). Data preprocessing involved the use of Cutadapt (v.2.6) and readfq (v1.0) to filter raw sequences, while Fast Length Adjustment of Short reads (v1.2.11) was implemented for consensus sequence generation. The denoising method of divisive Amplicon Denoising Algorithm was used to obtain amplicon sequence variants, after which the feature table was acquired. Taxonomic annotation was conducted by comparing representative operational taxonomic unit (OTU) sequences with the database, using an RDP classifier (v2.2). Diversity analysis, function prediction, and additional data analysis were based on OTU and taxonomic annotation. In this context, the P10 group samples were selected for 16S rRNA gene sequencing, denoted as the pomace group (P) in Section 3.4.

### 2.9. Statistical Analysis

Data were expressed as mean ± standard deviation (SD) or standard error (SEM). One-way analysis of variance (ANOVA) and a post hoc Tukey test were employed to evaluate significant differences (* *p* < 0.05, ** *p* < 0.01, *** *p* < 0.001), using SPSS (Version 25). The graphical representation of the data was conducted using GraphPad Prism 8.0.

## 3. Results

### 3.1. LCP-Ameliorated Colitis Symptoms

LCP was characterized by a rich content of dietary fiber (approximately 42.60%) and carbohydrates (around 40.29%), with the former primarily comprising insoluble fiber, constituting 68.81% of total dietary fiber (Table 3). The soluble sugar content was quantified as 18.44 g/100 g, wherein glucose and fructose represented 54.63% and 45.37%, respectively. The total polyphenol content was 9 g/100 g (dry weight). Following a one-week acclimatization period, the LCP diet was introduced, and DSS was administered during the final week to establish the chemically induced IBD mouse model (Figure 1A). There was no observed mortality among the mice during the experiment. The results showed that the LCP diet could significantly prevent weight loss (Figure 1B), improve appetite (Figure 1C), and reduce colon shortening (Figure 1F) caused by DSS administration. The disease activity index (DAI), a substantial marker for assessing the severity of intestinal inflammation, is evaluated by parameters including weight loss, stool consistency, and bloody stool fraction [33]. As colitis progressed, the DAI of the M group ascended gradually from the initial to the final day of DSS administration, whereas for the group treated with LCP this increase significantly decelerated. Furthermore, the DAI of the P10 group was consistently inferior to that of the P5 group from day 4 to day 7 (Figure 1E), indicating a milder disease manifestation in the P10 group. Representative images of the mice’s anus and colon showed that LCP mitigated symptoms of rectal bleeding and colon shortening (Figure 1H). The health status of the mice will be indirectly reflected in the organ index. Following DSS administration, the kidney index manifested a significant enhancement, a change the P10 diet ameliorated to control level. DSS administration also increased the liver and spleen indexes, although the P5 and P10 diets did not significantly restore these changes. (Figure 1G). Taken together, these results suggested that LCP notably alleviated colitis symptoms, with the P10 group demonstrating a more pronounced effect relative to the P5 group.

### 3.2. LCP Restored Damage to Colon Tissues and Modulated the Inflammatory Response Associated with Colitis

The histological examination of the intestinal tract via H&E staining showed enhanced preservations of mucosal structure and intestinal crypt in mice fed with LCP, accompanied by reduced inflammatory cell penetration in the mucosal, submucosal, and muscular layers, in comparison to the M group (Figure 2A,B). MPO activity, a marker of neutrophil infiltration, showed a 2-fold increase in the colon tissue after DSS administration (Figure 2D). The LCP diet decreased MPO activity to the level of the C group when compared with the M group. Colitis induced an upsurge in the expression of inflammatory cytokines in colon tissue, thereby precipitating systemic inflammation. The serum concentration of TNF-α in colitis mice was elevated, potentially mediated by the upregulation of *TNF-α* gene expression in colon tissue (Figure 2C,E). This upregulation was reversed by the LCP diet. Besides *TNF-α*, DSS also upregulated *IL-1β* and *IL-6* gene expressions, which were attenuated to levels approaching those in the C group by the LCP diet (Figure 2F,G). To summarize, the LCP diet demonstrated a capacity to inhibit colitis development via diminishing MPO activity and suppressing the expressions of inflammatory cytokines.

### 3.3. Protective Effects of LCP on the Intestinal Barrier

To assess the potential beneficial impact of the LCP diet on the conservation of the intestinal barrier, the mucin of colon tissue sections was initially stained with AB-PAS. Compared with the C group, the mucin staining area in the M group was reduced 8.3-fold, whereas the staining area in the P5 and P10 groups was restored to normality (Figure 3A,B). *Muc2* gene expression was markedly elevated in the P10 group compared to the M group (Figure 3D). Goblet cells, which secrete mucus in the intestinal crypts to form a protective mucus barrier, were counted, revealing a 7.2-fold decrease in the colitis mice compared to healthy mice. Conversely, the P5 and P10 groups restored the depletion of goblet cells 3.5- and 6.7-fold, respectively (Figure 3C). Tight junctions (TJs), comprising claudins, occludin, and zonula occluden (ZO) proteins, constitute an essential mechanical barrier within the intestinal tract [35]. In colon tissue, the gene expressions of *Claudin-1*, *Occludin* and *ZO-1* proteins were considerably suppressed in colitis mice but markedly increased in the P10 group (Figure 3E–G).

### 3.4. Modulation of LCP on Gut Microbiota

Here, we selected the samples of the P10 group for 16S rRNA gene sequencing. A species accumulation curve, which was delineated based on the number of samples and species OTUs, demonstrated no substantial increment in species as the sample quantity expanded, thereby indicating that the sample volume was adequate for reflecting species richness (Figure 4A). A Venn diagram visually displayed the overlay of OTUs among groups, revealing 343, 37, and 79 various microbes in the C, M, and P groups, respectively, with 48 overlapping microbes observed among them (Figure 4B). The M group displayed a diminished OTU count relative to the C and P groups. Both community richness and diversity indexes, as determined by Chao1, Ace, Simpson, and Shannon, were included in the α-diversity analysis. The results revealed significant suppressions in the richness and diversity of the M group. Conversely, the LCP diet distinctly ameliorated these metrics, exhibiting a diversity nearly analogous to the C group (Figure 4C–F). The results of Principal Coordinates Analysis (PCoA) and Non-metric Multidimensional Scaling (NMDS) revealed distinct clustering separations among the C, M, and P groups, manifesting disparate gut microbiota structures (Figure 4G,H). The UPGMA cluster tree showed that the C group and DSS administration groups (M and P groups) were located on distinct branches, thus possessing different microbial compositions. The M and P groups were relatively clustered, on two branches (Figure 4I).

Subsequently, we investigated the alternations in gut microbiota compositions consequent to the administration of DSS and the LCP diet. The most abundant phyla include Firmicutes, Bacteroidetes, Actinobacteria, Proteobacteria, and Verrucomicrobia. At the phylum level, Proteobacteria and Deferribacteres manifested a drastic elevation in colitis mice, augmented 10.7- and 3.5-fold compared to the healthy group, whereas Firmicutes, Actinobacteria, Bacteroidetes, and Tenericutes significantly dropped, 3.9-, 41.3-, 32.1-, and 150.4-fold (Figure 5A,B). In comparison to the M group, the LCP diet remarkably reduced Proteobacteria 2.3-fold, while increasing Bacteroidetes and Verrucomicrobia 83.7- and 5.5-fold. At the genus level, the abundance of *Escherichia*, *Enterococcus*, and *Clostridium* were 1334.7-, 558.6- and 3.3-fold higher in colitis mice than in healthy mice, but 2.9-, 136.8-, and 5.9-fold lower than in the P group (Figure 5H–J). *SMB53*, undetectable in the C group, was extremely elevated in colitis mice and significantly declined in the P group (Figure 5K). Following DSS administration, genera such as *Oscillospira*, *Coprococcus*, *Ruminococcus*, and *Bifidobacterium*, were decreased 2.8-, 456.9-, 20.2- and 7950.2-fold compared to the C group, a phenomenon partly reversed by the P group (Figure 5D–G). A reduction in *Mucispirillum* was observed in the P group compared to the M group, although without statistical significance. Interestingly, no substantial difference in *Akkermansia* between the C and M groups was observed; however, *Akkermansia* was 5.5- and 2.4-fold higher in the P group compared to the C and M groups (Figure 5C).

Linear discriminant analysis effect size (LEfSe) was employed to analyze microbial groups with significant effects across diverse groups. The histogram of linear discriminant analysis (LDA) illustrated biomarkers with statistical difference (Figure 6A,D). In the P group, the predominant bacteria were *Akkermansia*, *Parabacteroides*, *Roseburia*, *Anaeroplasma*, *Dorea*, *Odoribacter*, and *Turicibacter*, while in the M group the predominant bacteria were *Escherichia*, *Enterococcus*, and *SMB53*. Twenty-one pathways from Kyoto Encyclopedia of Genes and Genomes (KEGG), with varied levels of abundance, were observed between the C and M groups (eight increased and eleven decreased in the M group), including those associated with microbial genes, such as nucleotide metabolism, carbohydrate metabolism, vitamin metabolism, metabolism of cofactors, and amino acid metabolism (Figure 6B). Additionally, the LCP diet facilitated the restoration of the abundance of metabolic pathways, enhancing amino acid metabolism, cell growth and death, the digestive system, biosynthesis of other secondary metabolites, lipid metabolism, glycan biosynthesis and metabolism, and metabolism of terpenoids and polyketides, while decreasing infectious diseases (bacterial) (Figure 6C).

### 3.5. Effect of LCP on SCFAs in Intestinal Contents of Colitis Mice

SCFAs, produced by microbial fermentation, are posited to confer advantageous effects on colonic homeostasis. We quantified six SCFA levels in the intestinal contents of each group. The DSS administration manifested a decrease in all SCFA levels (Figure 7). Conversely, the LCP diet led to an increase in the SCFA levels in colitis mice, and, except for isobutyric acid, the levels of SCFAs in the P10 group reverted to those in the C group.

## 4. Discussion

The present study found that the LCP diet, characterized by a high content of dietary fibers and polyphenols, alleviated the symptoms of DSS-induced colitis. The LCP-mediated mitigative effects were probably attributed to the preservation of the intestinal barrier, attenuation of the inflammatory response, and modulation of gut microbiota. Consistent with previous investigations, the daily intake of berries or berry pomace has been demonstrated to alleviate colitis symptoms [16,30,31]. In the current work, we investigated the preventive effect of the LCP diet on colitis using a DSS-induced colitis model. We discovered that the symptoms of colitis, such as an increased DAI and reduced colon length, were significantly alleviated (Figure 1), while the natural anatomical integrity of the colon tissue was preserved in the P5 and P10 groups (Figure 2A). Intriguingly, the physical conditions of the mice in the P10 group were better than those in the model group, even though the P10 diet markedly increased the DSS consumption (Figure 1D), thus indicating a positive efficacy of 10% LCP in colitis alleviation.

In several inflammatory disorders, MPO has been proved to be a local mediator of tissue injury and subsequent inflammation [36]. The MPO activity was elevated nearly twofold in colitis mice compared to their healthy counterparts, yet it was markedly suppressed, by a factor of 2.3, in the P5 and P10 groups compared to colitis mice (Figure 2D), underscoring the considerable anti-inflammatory impact of LCP. Inflammatory cytokines are important in colitis pathogenesis, as they control the development of the inflammatory response [37]. We tested the genetic expression of inflammatory cytokines and found that the LCP diet reduced their expression in colitis, particularly in the P10 group, wherein levels of *TNF-α*, *IL-1β*, and *IL-6* were diminished 3.0-, 3.2-, and 2.1-fold compared to the M group (Figure 2E). It is reported that *L. caerulea* polyphenols, Cyanin 3-glucoside and (-)-epicatechin could reduce the production of multiple inflammatory cytokines under LPS-induced inflammation [38]. Inflammatory cytokine expression may be associated with SCFAs in the gut [39]. Acetic-, propionic-, butyric-, valeric-, isobutyric- and isovaleric-acid levels were 1.5-, 2.0, 2.1-, 4.6-, 1.7-, and 1.9-fold lower in the colitis model than in the C group (Figure 7). The SCFA levels in the P10 group were restored to those of healthy mice. This observation aligns with the trends of inflammatory cytokines and might partially be ascribed to the anti-inflammatory effects of SCFAs. SCFAs have been reported to regulate immune cell chemotaxis, reactive oxygen species, and cytokines, thus exerting anti-inflammatory actions [40,41,42]. For instance, butyrate substantially inhibited the gene expression of inflammatory cytokines and MPO production in neutrophils from IBD patients [43].

SCFAs are recognized as biomarkers reflecting alterations in gut microbial metabolism and microbiota composition in colitis models. Relative to the M group, the LCP diet fostered an enhancement in the abundance of SCFA-producing bacteria, such as *Akkermansia*, *Oscillospira*, *Coprococcus*, and *Bifidobacterium* (Figure 5C–E,G). Biomarkers for the P group, such as *Roseburia*, *Dorea*, and *Odoribacter* (Figure 6D), could generate SCFAs like acetic acid and butyric acid [40,44,45,46], to facilitate intestinal homeostasis. These might be associated with the increased SCFAs in the P5 and P10 groups. Recent research showed that *L. caerulea* administration increased the abundance of *Bifidobacterium* and *Akkermansia*, and modulated gut microbiota composition [47], which was consistent with our results. Additionally, other beneficial bacteria increased in the P group, such as *Coprococcus* and *Ruminococcus*, demonstrating negative correlations with disease in rectum samples [48], and were beneficial for reducing intestinal inflammation [49]. The dominant microorganism *Anaeroplasma* in the P group has been reported to increase the levels of the anti-inflammatory cytokine TGF-β and mucosal IgA, exerting anti-inflammatory effects [50]. In the meantime, unlike our results, previous studies have found that *Anaeroplasma* was enriched in mice with colitis-associated colorectal cancer compared to healthy controls [51,52]. The observed variation in the abundance of *Anaeroplasma* could be explained by various inflammatory niches due to different disease models.

On the other hand, SCFAs, energy sources for colonocytes, have positive effects on the intestinal barrier, which is crucial in IBD pathogenesis. In the present investigation, *Claudin-1* gene expression was upregulated 1.8- and 3.8-fold in the P5 and P10 groups (Figure 3E) in comparison to the M group, and this recovery was observed to escalate in correspondence with the dosage increase. However, the proteins *Occludin* and *ZO-1* exhibited remarkable elevations only within the P10 group, by factors of 4.1 and 2.4 (Figure 3F,G). A possible explanation for this observation might be the higher concentration of polyphenols and dietary fiber within the LCP in the P10 group. Research has demonstrated that an 8% inclusion of grape peel powder in the diet and *Citrus kawachiensis* peel powder [16,31] significantly ameliorated intestinal barrier function in colitis mice by restoring TJs, which was mostly due to the contribution of dietary fiber and polyphenols. In fat SD rats, *L. caerulea* polyphenol administration exhibited significantly higher protein-expression levels of occludin to enhance intestinal barrier function [23]. The enhanced barrier could also be the contribution of the increased SCFAs. Targeted butyrate release within the intestine could repair intestinal barrier damage in colitis mice [53]; in TNBS-induced mice, sodium butyrate remarkably alleviated intestinal epithelial barrier dysfunction, as evidenced by the enhancement in TJ protein expression [54].

The mucosal barrier, serving as the initial defense against microbial attack, involves goblet cells that synthesize highly glycosylated mucin, with Muc2 being the predominant mucin [55,56]. This research unveiled the fact that the LCP diet preserved the mucosal barrier by increasing the number of goblet cells and elevating both *Muc2* gene expression and mucin concentration. The mucus distribution and goblet cells in the P5 and P10 groups approximated those of the C group (Figure 3B,C). Compared with the M group, *Muc2* gene expression was slightly elevated in the P5 group and significantly elevated in the P10 group (Figure 3D). Interestingly, *Akkermansia*, known to proliferate within the mucus layer and metabolize host-secreted mucin, may play a role in intestinal tract colonization and pathogen protection via competitive exclusion [57]. In our investigation, *Akkermansia* was identified as a biomarker for the LCP-fed group and was greatly enhanced by the LCP diet in colitis mice, 5.5-fold. This could be attributed either to enhanced mucin secretion caused by the LCP diet, stimulating rapid *Akkermanisia* multiplication, or to the LCP diet-induced proliferation of *Akkermanisia*, in turn heightening mucin demand—both of which warrant subsequent exploration. The polyphenol content of LCP (9.00 g/100 g dw) was detected to be about 10 times higher than that of *L. caerulea* fruit juice powder (0.90 g/100 g dw). Previous reports showed that *L. caerulea* possesses a high polyphenol content, with pomace containing 4.3-fold more polyphenols than fresh berries [18,19]. A study showed that *L. caerulea* polyphenols improved intestinal barrier function by increasing fecal mucin in rats [58]. Supportive evidence indicated that proanthocyanidins in cranberries enhance mucosal barrier integrity by stimulating Muc2 secretion, thereby fostering *Akkermansia* proliferation through the establishment of a suitable environment [59]. Furthermore, natural dietary fiber blends might confer additional advantages, such as grape peel powder preventing apoptosis in the intestinal mucosa through GSH recycling and antioxidant enzymes [60]. The combined effect of the mucus layer and *Akkermanisia*’s competitive repulsion of pathogens conferred dual protection to the intestinal barrier.

The homeostasis of microbiota within animals has been previously determined to be subject to dietary influences [3,5,61]. In our study, LCP provided an abundance of dietary fiber and polyphenols, which remain undigested within the upper digestive tract, subsequently becoming substrates for the gut microbiota. Species richness and diversity typically dropped in the gut microbiota of IBD patients [3], whereas the LCP diet significantly recovered this trait and improved microbiota structure in colitis mice (Figure 4C–F). Further analysis revealed that following the LCP diet, the abundance of pathogenic bacteria in colitis mice declined at the genus level (Figure 5H–L). *Escherichia* and *Enterococcus* were considerably suppressed by LCP. Previous studies have implied that *Escherichia* and *Enterococcus* were connected to the pathogenesis of IBD and intestinal inflammation [62]. *L. caerulea* extract exerted an antibacterial effect on these microorganisms [18]. In contrast, within the M group, *Mucispirillum* demonstrated an increasing trend, while exhibiting a reduction in the P group. Although both *Mucispirillum* and *Akkermansia* metabolize mucin, the former was determined as a biomarker in animals with colitis, and was related to pro-inflammatory bacteria [63,64]. Regarding the prediction of KEGG functional pathways, the LCP diet enhanced metabolism of terpenoids and polyketides, the digestive system, and biosynthesis of other secondary metabolites, potentially indicative of an augmented capability for digestion and metabolism within the mice.

## 5. Conclusions

According to this research, daily LCP supplementation could alleviate the symptoms of DSS-induced colitis. The multiple mechanisms of relieving colitis included inhibiting the inflammatory cytokine expression, maintaining the colonic mucosal barrier and TJ integrity, and regulating the intestinal microbial composition, as well as recovering SCFA levels. These results demonstrated the functional role of LCP in alleviating colitis, which is helpful for guiding the future application of LCP. More research is needed to pinpoint the specific bioactive compounds present in LCP that were directly involved in the induction of the observed beneficial effects in the gastrointestinal tract.

## Figures and Tables

**Figure 1 foods-12-03329-f001:**
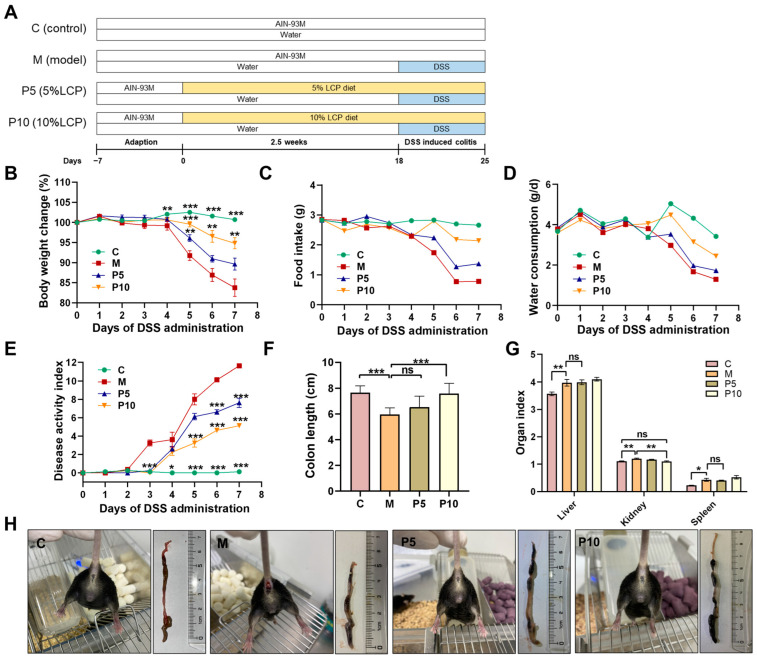
LCP-ameliorated colitis symptoms. (**A**) Animal feeding scheme. (**B**) Body-weight change. (**C**) Food intake. (**D**) Water consumption. (**E**) Disease activity index (DAI). (**F**) Colon length. (**G**) Organ index. (**H**) Representative images of the mice’s anus and colon. Data are presented as mean ± SEM (*n* = 8). One-way ANOVA with post hoc Tukey test; “ns” denotes comparisons without significance. The asterisks marked in the B and E diagrams indicate the significant differences compared with the M group. * *p* < 0.05, ** *p* < 0.01, *** *p* < 0.001.

**Figure 2 foods-12-03329-f002:**
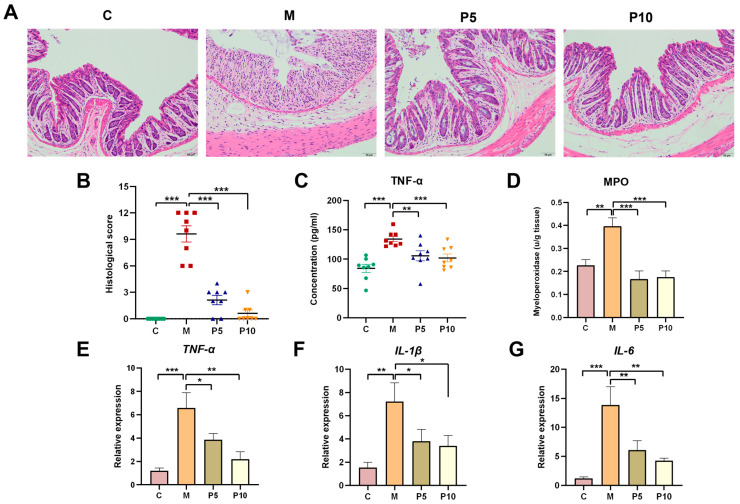
LCP regulated the inflammatory response. (**A**) Histological examination, Scale bars, 50 μm. (**B**) Histological score of colon section. (**C**) Concentration of TNF-α in serum. (**D**) Myeloperoxidase (MPO) activity in colon tissues. (**E**–**G**) Gene expressions of inflammatory cytokine in colon tissues. Data are presented as mean ± SEM (*n* = 8). One-way ANOVA with post hoc Tukey test. * *p* < 0.05, ** *p* < 0.01, *** *p* < 0.001.

**Figure 3 foods-12-03329-f003:**
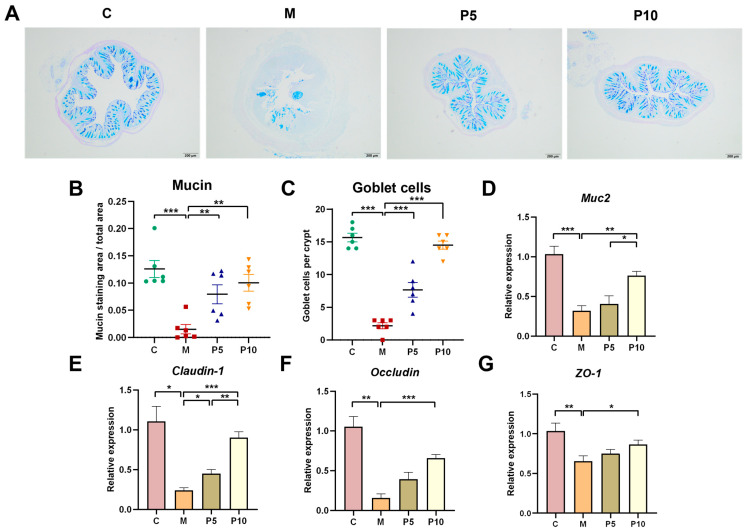
LCP maintained the colonic mucosal barrier and Tight junction (TJ) integrity. (**A**) Representative images of mucin-stained colon tissue sections. (**B**) Mucin staining area relative to total cross-sectional area of the colon. (**C**) The number of goblet cells in each crypt. (**D**) Gene expression of *Muc2* in colon tissue. (**E**–**G**) Gene expressions of TJ proteins in colon tissues. Data are presented as mean ± SEM (n = 6–8). One-way ANOVA with post hoc Tukey test. * *p* < 0.05, ** *p* < 0.01, *** *p* < 0.001.

**Figure 4 foods-12-03329-f004:**
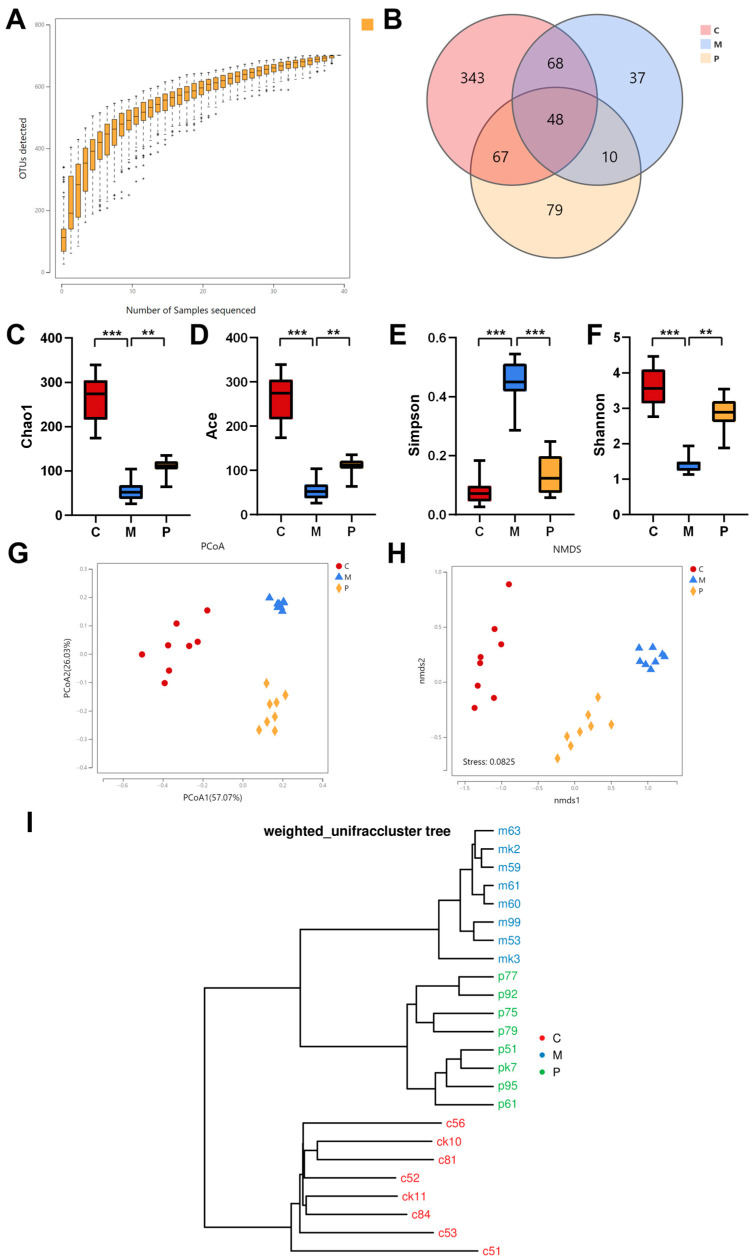
LCP-regulated intestinal microbial diversity in colitis mice. (**A**) Species accumulation curve. (**B**) Venn diagram of operational taxonomic unit (OTU) composition and overlap. (**C**–**F**) α-diversity index. The interquartile range (IQR) and median values are displayed within boxes, with whiskers extending to 1.5 times the IQR (*n* = 8). One-way ANOVA with post hoc Tukey test. ** *p* < 0.01, *** *p* < 0.001. (**G**) β-diversity assessed by PCoA based on weighted UniFrac distances. (**H**) β-diversity assessed by NMDS (Stress < 0.0825). (**I**) β-diversity assessed by cluster tree.

**Figure 5 foods-12-03329-f005:**
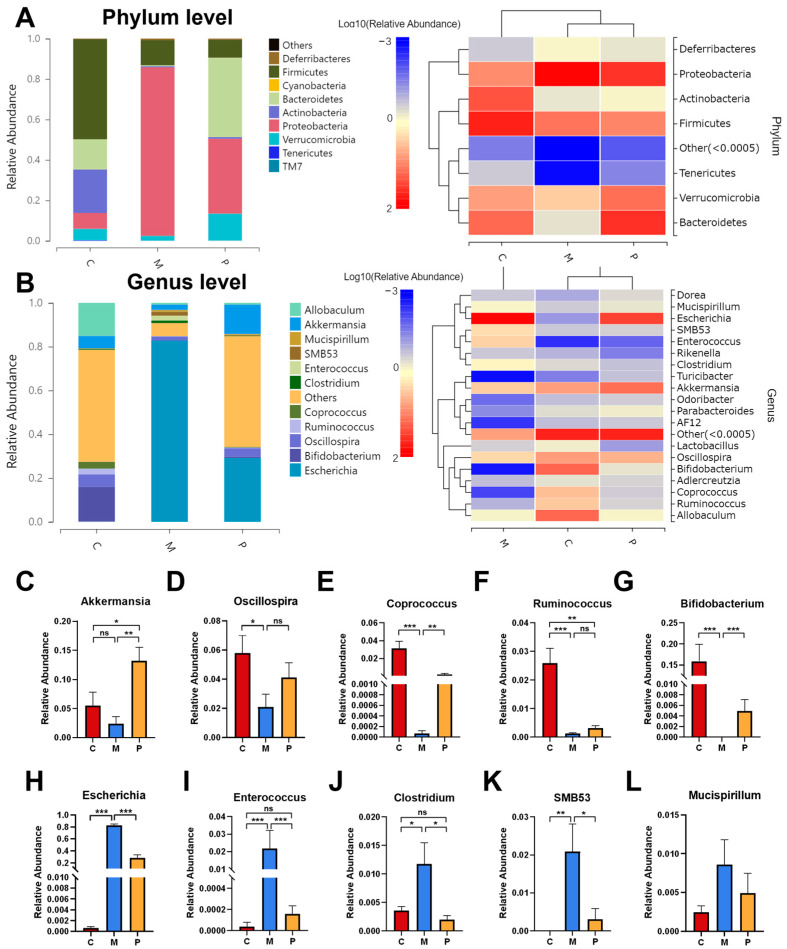
Modulation of LCP on gut microbiota composition. (**A**) Taxonomic distributions of the relative abundance and community heatmap at the phylum level. (**B**) Taxonomic distributions of the relative abundance and community heatmap at the genus level. (**C**–**G**) Relative abundance of species displaying significant reduction in colitis mice. (**H**–**L**) Relative abundance of species demonstrating significant elevation in colitis mice. Data were presented as mean ± SEM (*n* = 8). One-way ANOVA with post hoc Tukey test. * *p* < 0.05, ** *p* < 0.01, *** *p* < 0.001. “ns” denotes comparisons without significance.

**Figure 6 foods-12-03329-f006:**
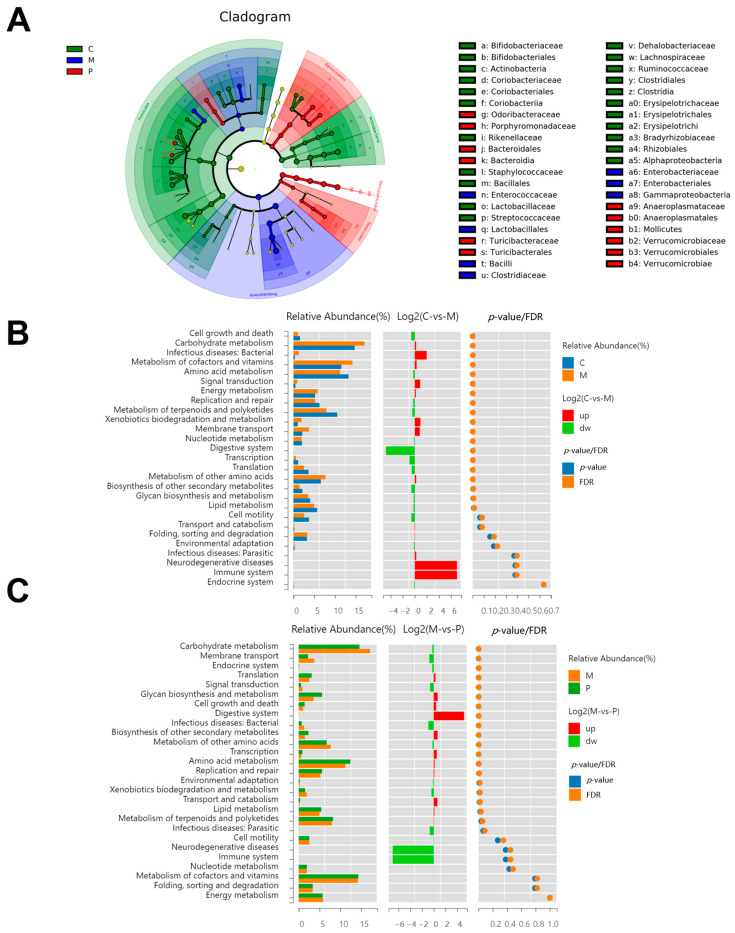
Effects of LCP on dominant microorganisms and functional difference analysis. (**A**) Taxonomic cladogram. (**B**,**C**) Differential pathways among the C, M and P groups using two-sided Wilcox test at *p* < 0.05. (**D**) Linear discriminant analysis (LDA) histogram with the score above 2.

**Figure 7 foods-12-03329-f007:**
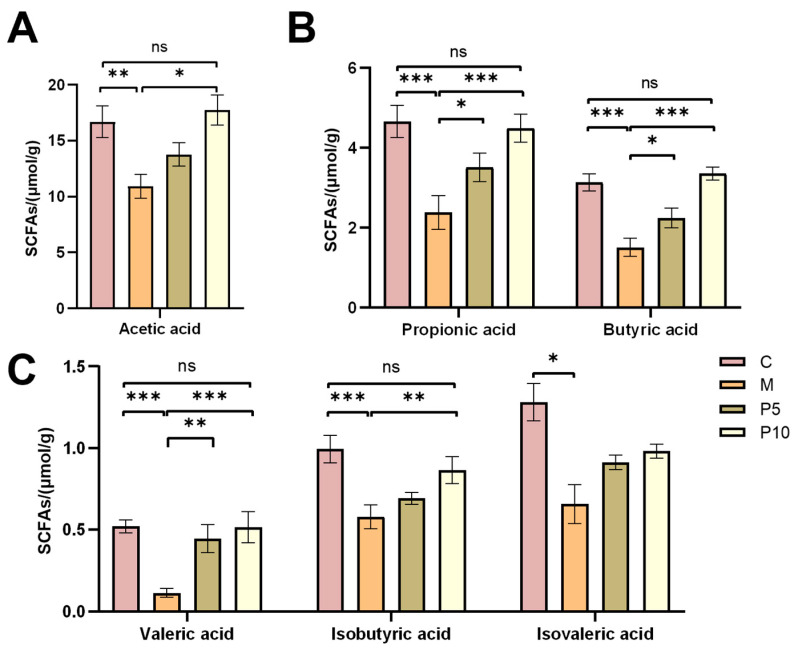
Effects of LCP on short-chain fatty acids (SCFAs) in colitis mice. (**A**–**C**) The levels of acetic acid, propionic acid, butyric acid, valeric acid, isobutyric acid, and isovaleric acid in the intestinal contents of each group. Data are presented as mean ± SEM (*n* = 8). One-way ANOVA with post hoc Tukey test. * *p* < 0.05, ** *p* < 0.01, *** *p* < 0.001. “ns” denotes comparisons without significance.

**Table 1 foods-12-03329-t001:** Experimental diet ingredients and composition.

	AIN-93M		5% LCP Diet (P5)		10% LCP Diet (P10)	
	g	kcal	g	kcal	g	kcal
Casein	140	560	135.5	542	130.9	523.6
Corn starch	465.69	1862.77	449.7	1798.8	433.2	1732.8
Maltodextrin	155	620	155	620	155	620
Sucrose	100	400	100	400	100	400
Soybean oil	40	360	38.7	348.3	37.35	336.15
Cellulose	50	-	50	-	50	-
Mineral	35	-	35	-	35	-
Vitamin	10	-	10	-	10	-
L-cysteine	1.8	-	1.8	-	1.8	-
Choline Bitartrate	2.5	-	2.5	-	2.5	-
TBHQ	0.008	-	0.008	-	0.008	-
LCP	-	-	52	113.41	106	231.19
Total	1000	3802.77	1030.21	3822.51	1061.76	3843.74
Calorie (%)						
Lipids	9.47		9.42		9.36	
Protein	14.73		14.60		14.47	
Carbohydrates	75.81		75.21		74.60	

TBHQ: tert-Butylhydroquinone is an antioxidant. LCP: *Lonicera caerulea* pomace.

**Table 2 foods-12-03329-t002:** Primer sequences of RT-qPCR.

Gene	Sequences (5′ to 3′)	
	Forward Primer	Reverse Primer
*β-actin*	TGTCCACCTTCCAGCAGATGT	AGCTCAGTAACAGTCCGCCTAGA
*TNF-α*	TGCTTTCTGTGCTCATGGTG	GACTAGCCAGGAGGGAGAAC
*IL-1β*	ACTCATTGTGGCTGTGGAGA	AGCCTGTAGTGCAGTTGTCT
*IL-6*	ACTTCACAAGTCCGGAGAGG	TGCAAGTGCATCATCGTTGT
*Muc2*	ACGTGTCATATTTGCACCTCT	TCAACATTGAGAGTGCCAACT
*Claudin-1*	ACGGTCTTTGCACTTTGGTC	GGGAGAGGAGAAGCACAGTT
*Occludin*	TGCGGTGACTTCTCCAAACT	GGGGAACGTGGCCGATAT
*ZO-1*	TCTTCCATCATTTCGCTGTGT	TCTGAAACCATCAAGTCCACA

**Table 3 foods-12-03329-t003:** Chemical composition of *L. caerulea* pomace powder.

Composition (g/100 g of Dry Weight)	LCP (%)
Ash	1.95 ± 0.02 ^1^
Lipids	3.73 ± 0.06
Protein	11.43 ± 0.13
Carbohydrates	40.29 ± 0.18
Soluble sugar	18.44 ± 0.13
Fructose	8.37 ± 0.04
Glucose	10.07 ± 0.12
Sucrose	ND ^2^
Maltose	ND
Dietary fiber	42.60 ± 0.18
Soluble fiber	13.29 ± 0.08
Insoluble fiber	29.31 ± 0.13
Total polyphenol	9.00 ± 0.07

^1^ Results are presented as mean ± SD. ^2^ ND means not detected.

## Data Availability

The data presented in the study are deposited in the National Center for Biotechnology Information (NCBI) repository, accession number is PRJNA947371.

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
