# Peer review of "Lonicera caerulea Pomace Alleviates DSS-Induced Colitis via Intestinal Barrier Improvement and Gut Microbiota Modulation"

_foods, 2023, doi:10.3390/foods12183329_

Round 1

Reviewer 1 Report

Dear authors,

Thank you for your paper. I quite enjoyed reading the results and the discussion of this paper. However I feel the introduction (editing-wise) and the methodology (a lot of detail is missing) can improved upon. Below are my comments on your paper:

Introduction

Line 28 – ‘Ulcerative’ should be ‘ulcerative (small u)’

Line 32 – considered (add that) the pathophysiology involves gene inheritance, immune imbalance,

Line 33 – and host reaction resulted (replace with resulting) from exogenous substances…

Line 34 – diarrhea, (add and) mucus discharge…

Line 38 – therapeutic strategies that utilizing (replace with utilize) natural materials

Line 39 – 40 – this sentence is a bit confusing, the authors need to clarify if they mean diet can reduce the occurrence of colitis 6.3 fold or diet can increase colitis up to 6.3 fold. The use of the word ‘survival’ creates confusion.

Line 43 – 45, please rewrite for grammatical correctness.

Line 49 – L. caerulea, also familiar (replace with known) as honeysuckle berry or haskap

Line 53 – as a traditional food from a third (add world) country

Section 2.4 in brackets Elisa should be in all caps (ELISA)

Methodology

Section 2.4 and 2.5 – more detail is needed as to how the experiment was conducted – simply stating according to the kit/conducted by a company is not sufficient.

Section 2.6 – detail how was the RTPCR was conducted – details on various times and conditions for denaturation, annealing and elongation etc. are necessary.

Section 2.7 – details of gas chromatography steps are needed – what were samples dissolved in? vapourisation? Separation? Detection?

Results

Line 171 – The total polyphenol was 9,000.41 mg/100 g – all other data is reported as g/100g – this should be the same for TPC = 9 g/100 g and this data should be added to table 3.

Line 172 – After one-week acclimatation (replace with acclimatization).

Line 176 – DAI should be written out, first use after abstract.

Line 184 – 188 – What to the authors mean by ‘increased kidney index’, ‘increased liver index’ or ‘increased spleen index’?

Line 185 – 187 – The authors state that DSS increase the liver and spleen index compared to the control (which was significant according to graph 1G) however from the graph it is clear that P5 and P10 are also significant – and the authors state this is not significant – please clarify/elaborate.

Figure 3 – Label graph B and C – as mucin and Goblet cells, respectively.

Figure 4 – Venn diagram B, needs to be explained a bit more in the legend. In legend all it says is Venn diagram, it needs more detail. In text only the numbers 343, 37 and 79 are explained, what about the rest?

Discussion

Line 335-336 – We tested at (remove at) the genetic expression of inflammatory cytokine(s) and found that the LCP diet…

Line 361 – what are these harmful reports of Anaeroplasma? To balance this sentence and not only rely on the beneficial effects, but some of the negative effects should also be reported.

Line 390 and 391, convert to g/100 g as most of the data in this paper is in g/100 g.

Dear authors,

This paper although very interesting, in some instances was difficult to read due to the English editing - please edit the paper to read grammatically correct - I have given some examples for the introduction.

Reviewer 2 Report

The article evaluates the alleviative effects of Lonicera caerulea on Dextran Sulfate Sodium (DSS)-induced colitis. In order to determine this effect, a comprehensive study was conducted that also investigated the underlying mechanisms. The article is written fluently and the results are adequately discussed.

Please, take in mind the following recommendations:

 My biggest criticism is that the graphics are not clear. Their resolution can be increased or their size can be changed.

In the article, Lonicera caerulea is sometimes written as an abbreviation, sometimes the full name is written, and sometimes the genus name is abbreviated. The spelling should be corrected throughout the article.

The Conclusions section is imprecise and generic. A very brief summary of all acquired results and subsequent conclusions is expected.

Minor editing of English language required

Reviewer 3 Report

The manuscript by Baixi Zhang and co-workers entitled ”Lonicera caerulea pomace alleviates DSS-induced colitis via intestinal barrier improvement and gut microbiota modulation” presents a very interesting study in which the Authors used a by-product (pomace) obtained during blue honeysuckle juice production to investigate the potential use of L. caerulea pomace (LCP) as a dietary supplement, and demonstrated a protective effect in the intestines. The Authors based their study design on results obtained by another research group, who showed that peel-based dried pomace of L. caerulea contained more polyphenolic compounds than the pulp, thus should exert strong antioxidative and anti-inflammatory activities in the GI tract.  To test this hypothesis the Authors used a in vivo mouse model of dextran sulfate sodium-induced colitis. Prior colitis induction with DSS mice were already fed a diet supplemented with 5% or 10% addition of LCP for 18 days, and then animals were subjected to colitis induction for the following 7 days, during which the dietary intervention was continued. This model enabled to test preventive not therapeutic effect of LCP consumption; however, the results obtained are very promising.  In general, the study was well planned and the methods were chosen correctly. The Authors obtained many valuable data, but the manuscript still needs several corrections. The detailed remarks are listed below:

·        Table 1 presents the composition of diets used in the study. Why did the mass [g] and food energy [kcal] differed slightly among the three diets (control: AIN-93M and P5 and P10)? In addition, the Authors should explain the symbol: TBHQ in the table caption.

·        Table 2 presents the chemical composition of LCP powder. It would be better to make a defined cut-off points/lines which divide the main groups of compounds and their subgroups, e.g. soluble sugar (subgroup: fructose, glucose, sucrose, maltose), dietary fiber (subgroup: soluble fiber, insoluble fiber). This would also match the description in the main text (lines: 168-188). In addition, there is not information about the concentration of polyphenols in Table 3, although the Authors present the values in the text (line: 171 and 390-391). This should be added to Table 3.

·        Did the Authors analyze in more details the composition of polyphenols in LCP?

·        In Figure 1 all data are too small. Readers need to magnify this figure to see Fig.1.A  – the scheme of in vivo experiment (I used 150% magnification) and Fig.1H – images of mice colon. Fonts in axes and legends of graphs Fig.1B-G are too small.

·        Figure 1H  does not present paragraphs of mice anus and colon, but micrographs or images – this should be corrected in line: 182 and Figure 1 caption.

·        Figure 2A – it seems that the images of tissues in groups C and P5 are taken at is different magnification than those representing groups M and P10. This should be unified.  

·        In section 3.4. Modulation of LCP on gut microbiota the abbreviation: OUT should be explain when it is used the first time, for readers who are not familiar with units used in microbial taxonomy. Furthermore, in this section the Authors compare the microbiota composition among three groups: C, M and P – which P was used in this comparison: P5 or P10, or maybe summarized results from both experimental treatments?

·        Figure 4: In graphs 4G and 4H the fonts are too small; figure 4I is generally too small.

·        Figure 5A and 5B - the font in the title of graphs can be smaller but the font of the graph legends should be enlarged.

·        Figure  6 is generally too small. Maybe it would be better to divide the results presented into two figures (Figure 6B could be presented separately, which would allow its enlargement). A potential reader is not able to read any information from this figure even after 300% magnification of the text.

·        Figure 7 should also be enlarged. Graphs 7A and 7B could be next to each other, and graph 7C could be shown below, as the last three SCFAs share a common scale on the Y axis.

·        The Authors should avoid expressions stating that berry pomaces improve colitis, because this may be interpreted that these fruits enhance these symptoms. Such expression was used in several places in the text: lines: 322, 324-325, 427-428. This expression can be replaced by: “LCP alleviates the symptoms of colitis”, “LCP reduce the symptoms of colitis”, etc.

·        Sentence in lines: 358-361: “The dominant microorganism Anaeroplasma has  been claimed that it increased the levels of the anti-inflammatory cytokine TGF-β and mucosal IgA, suggesting that it may possess anti-inflammatory properties [43], but there are also harmful reports in other articles [44].” should be rephrased, because it is hard to understand the content and sense of it.

·        Sentence in lines: 362-362: “On the other hand, SCFAs, major energy sources of colonocytes, have positive effects in intestinal barrier, which is crucial in IBD pathogenesis.” should be changed.

Reviewer’s suggestion: “…energy sources for colonocytes..”

·        Generally, in the discussion section the Authors should focus more on comparing and relating their results with other studies on the effects of L. caerulea. When reading the present version of this section the reader may get an impression that the effects observed in the study where only connected with the presence of dietary fiber and polyphenols in the dietary supplement, but the source of these compounds is irrelevant. The Authors compared their results with results of studies conducted on cranberries, lychee pulp, grape peel powder, but did not underline or prove the value of  L. caerulea and its unique composition.

·        What does it mean: “digestive pathway of mice”? Rephrase the sentence in lines: 416-419.

·        Rephrase sentences in lines: 420-423, because they don’t make sense (“Increased amino acid metabolism may adjust the equilibrium of proteins and calories, as well as the growth of gut microbiota [58]. The down-regulation of infectious diseases pathway may imply that LCP is resistant to pathogens and diseases.”)

·        Rephrase the last sentence of the discussion section (lines: 423-425): “To sum up, the LCP diet modulated the gut microbiota of colitis and alleviated the symptoms, but more research is needed to pinpoint the exact bioactive fractions that are at work and how they affect the gut microbiota and metabolites in colitis.

Reviewer’s suggestion: To sum up, the LCP diet modulated the gut microbiota in colitis and alleviated the symptoms, but more research is needed to pinpoint the specific bioactive compounds present in LCP that were directly involved in induction of the observed beneficial effects in the gastrointestinal tract.

·        Line 431: replace “microbial structure” with “microbial composition”

·        Rephrase sentences in lines: 433-436, because they don’t make sense (for example change: “this research not only developed the functional effects…”, to: “this research not only demonstrated the functional effects…”.)

The manuscript needs thorough language editing, because there are many grammar and linguistic mistakes.
